# Identification of Quantitative Trait Loci for Node Number, Pod Number, and Seed Number in Soybean

**DOI:** 10.3390/ijms26052300

**Published:** 2025-03-05

**Authors:** Chunlei Zhang, Bire Zha, Rongqiang Yuan, Kezhen Zhao, Jianqiang Sun, Xiulin Liu, Xueyang Wang, Fengyi Zhang, Bixian Zhang, Sobhi F. Lamlom, Honglei Ren, Lijuan Qiu

**Affiliations:** 1Soybean Research Institute of Heilongjiang Academy of Agriculture Sciences, Harbin 150086, China; zhangchunlei1989@yeah.net (C.Z.); 15899078694@163.com (B.Z.); yrq18846121189@163.com (R.Y.); zhaokz928@163.com (K.Z.); liuxiulin1002@163.com (X.L.); hljsnkywxy@163.com (X.W.); ddszhangfy2019@163.com (F.Z.); 2College of Modern Agriculture and Ecological Environment, Heilongjiang University, Harbin 150080, China; 3College of Agronomy, Shenyang Agricultural University, Shenyang 110065, China; haassjq@163.com; 4Institute of Crop Sciences, Chinese Academy of Agricultural Sciences, National Key Facility for Crop Gene Resources and Genetic Improvement (NFCRl), Ministry of Agriculture and Rural Affairs, Key Laboratory of Crop Gene Resource and Germplasm Enhancement, Ministry of Agriculture and Rural Affairs, Beijing 100081, China; 5Institute of Biotechnology, Heilongjiang Academy of Agricultural Sciences, Harbin 150086, China; hljsnkyzbx@163.com; 6Work Station of Science and Technique for Post-Doctoral in Sugar Beet Institute, Heilongjiang University, 74 Xuefu Road, Harbin 150000, China; sobhifaid@alexu.edu.eg; 7Plant Production Department, Faculty of Agriculture Saba Basha, Alexandria University, Alexandria 21531, Egypt

**Keywords:** soybean, differential expression analysis, SLAF sequencing, QTL mapping

## Abstract

Optimizing soybean yield remains a crucial challenge in meeting global food security demands. In this study, we report a comprehensive genetic analysis of yield-related traits in soybeans using a recombinant inbred line (RIL) population derived from crosses between ‘Qihuang 34’ (GH34) and ‘Dongsheng 16′ (DS16). Phenotypic analysis across two years (2023–2024) revealed significant variations between parental lines. Through high-density genetic mapping with 6297 SLAF markers spanning 2945.26 cM across 20 chromosomes, we constructed a genetic map with an average marker distance of 0.47 cM and 99.17% of gaps under 5 cM. QTL analysis identified ten significant loci across both years: in 2023, we detected six QTLs, including a major main stem node number (MSNN) QTL on chromosome 19 (LOD = 22.59, PVE = 24.57%), two seed number (SN) QTLs on chromosomes 14 and 18 (LOD = 2.52–2.85, PVE = 7.35% combined), and one pod number (PN) QTL on chromosome 20 (LOD = 4.68, PVE = 5.85%). The 2024 analysis revealed four major QTLs, notably a cluster on chromosome 19 harboring significant loci for MSNN (LOD = 37.92, PVE = 43.59%), PN (LOD = 18.16, PVE = 23.02%), and SN (LOD = 15.24, PVE = 19.59%). Within the stable chromosome 19 region, we identified seventeen candidate genes involved in crucial developmental processes. Gene expression analysis revealed distinct temporal patterns between parental lines during vegetative and reproductive stages, with GH34 showing dramatically higher expression of key reproductive genes *Glyma.19G201300* and *Glyma.19G201400* during the R1 stage. Our findings provide new insights into the genetic architecture of soybean stem node development and yield components, offering multiple promising targets for molecular breeding programs aimed at crop improvement.

## 1. Introduction

The soybean (*Glycine max* (L.) Merr.) is a vital agricultural crop worldwide, providing an essential source of protein and oil for human consumption, animal feed, and diverse industrial uses [1,2]. With the rising demand driven by population growth and dietary shifts, it is essential to improve the production and adaptability of soybeans [3]. China stands as the world’s largest consumer of soybeans, with its commercial products increasingly reliant on soybean imports [4]. Over the past fifty years, efforts to enhance soybean yields in China have seen minimal progress highlighting an urgent need for the country to boost domestic production and achieve self-sufficiency in soybean cultivation [5]. Breeders are focusing on various yield-related traits to enhance soybean production, and one key trait is the stem node, which influences the plant’s architecture, adaptability, and yield potential [6,7].

Several QTLs linked to yield components have been identified in previous studies [8,9]. Researchers have discovered numerous QTLs associated with yield traits, particularly seed weight [10]. However, there has been limited research examining the features of main stems, pods, and seed numbers. One of the significant qualities in soybean breeding is the node number, a crucial characteristic affecting plant architecture, total biomass accumulation, and, ultimately, yield potential [11]. The architecture of the soybean plant, especially its node configuration, is crucial for photosynthesis and nutrient distribution, which directly influences seed yield [6,9]. The MSNN is a vital factor influencing plant height and branching, impacting light interception and, therefore, the plant’s capacity for effective photosynthesis [7]. Furthermore, node number can affect the allocation of reproductive structures, including pods and seeds, thus directly impacting yield results [12,13]. Consequently, comprehending the genetic foundation of node number variability in soybean populations is crucial for the creation of high-yield cultivars [13,14]. MSNN is associated with several key agronomic traits, including plant height, days to flowering, and days to maturity. However, MSNN is a complex quantitative trait controlled by multiple genes and is affected by environmental factors as well as genotype–environment interactions [15,16]. These genes can exert varying effects, ranging from minor to substantial, and may participate in various molecular and biological processes, functioning together within a network that can behave differently under diverse geographic and environmental conditions [17]. The seed number is determined by three factors: plant density, number of pods per plant (NPP), and number of seeds per plant (NSP). Both NPP and NSP are influenced by branch number and main stem number, though the genetic basis of these characteristics remains poorly understood [10,18,19]. PN is a crucial criterion in soybean classification and shows stability across years and environments [20]. Recent evidence suggests that one-seeded pods are controlled by two of three recessive genes. However, molecular studies on pod and seed number traits and their relationships remain limited [21]. Research in legumes has provided valuable insights into pod and seed number regulation. In soybeans, the map-based cloning of the Ln gene revealed that *Gm-JAG1* controls both leaf morphology and pod type. A specific amino acid change in *Gm-JAG1′s ERF* motif increased multiple-seeded pod numbers, leading to higher NSP and overall seed yield [22,23]. Another soybean gene, *GmCYP78A10b*, was found to increase seed size while reducing NPP, though this trade-off did not affect overall plant yield [24]. Similar genetic studies in other legumes have identified various mechanisms controlling pod and seed traits. These include a pentatricopeptide repeat (PPR) gene affecting NPP in chickpeas, simple sequence repeats (SSR) markers linked to NSP in common beans, and the relationship between non-structural polysaccharides and yield components in broad beans [25,26]. Researchers have also found co-localized QTLs for productivity, flowering, and seed development across multiple legume species. Recent metabolic studies have furthered our understanding of pod development [26]. In soybeans, eight metabolic pathways were found to contribute to the development of pods with four or more seeds [27,28,29]. In rapeseed, researchers identified both pleiotropic QTLs affecting NSP and seed yield, as well as a strong genetic linkage between NPP and NSP traits [30]. To better understand the genetic relationships among yield components in soybean, researchers have employed multivariable conditional analysis techniques. This approach has successfully revealed genetic associations among multiple phenotypes at the QTL level. For example, conditional QTL mapping in soybeans has shown that pod length has a greater influence on 100-pod weight than pod width, and that branch number significantly affects resistance to late leaf spot [21,31]. These findings suggest that similar approaches could be valuable for analyzing genetic relationships among PN and SN. Single nucleotide polymorphisms (SNPs) are important for making high-density genetic linkage maps and mapping quantitative trait loci (QTL) [28,32,33,34,35,36]. Many researchers indicated that using both linkage analysis and association analysis together is more accurate and useful than using only one of them [37,38,39]. Linkage analysis usually only looks at populations that come from two parental lines. This limits genetic diversity and, in turn, its ability to find things [40]. SLAF-seq is an innovative method that facilitates the swift development of SNP markers following the building of an SLAF-seq library [41]. SLAF-seq technology has been effectively utilized to create high-density maps of various species owing to its benefits of high throughput, precision, cost-effectiveness, and rapid cycle times [42,43,44]. This study aimed to genetically map QTLs associated with three yield-related traits using the Dongsheng 16 and Qihuang 34 F2-6 RIL population, while also identifying candidate genes involved in soybean MSNN, PN, and SN. We utilized a genetic linkage map obtained from SLAF sequencing to identify candidate intervals for the primary loci associated with MSNN. This research aimed to identify and characterize QTLs linked to main stem nodes in the soybean recombinant inbred lines (RILs) population, thereby enhancing the effectiveness of breeding programs.

## 2. Results

### 2.1. Phenotypic Characteristics of Parents and RILs

The phenotypic analysis of PN, SN, and MSNN revealed significant variations between genotypes DS16 and QH34 across two years (2023 (Harbin) and 2024 (Sanya)) (Figure 1A,B). In 2023, substantial differences were observed between the two genotypes. QH34 demonstrated consistently higher values across all traits compared to DS16. PN in QH34 (77.0 ± 2.5) was significantly higher than DS16 (45.20 ± 1.16). Similarly, SN showed marked differences with QH34 (156.00 ± 12.65) exhibiting higher values than DS16 (96.40 ± 4.33). MSNN also followed this trend, with QH34 (22.5 ± 0.9) showing higher values compared to DS16 (16.80 ± 0.86) (Figure 1A). The 2024 data showed an overall decrease in trait values compared to 2023, though genotypic differences persisted. PN maintained a highly significant difference between QH34 (38.0 ± 1.91) and DS16 (24.60 ± 1.16). SN showed no significant difference between genotypes, with values of approximately 69.02 ± 10.39 for QH34 and 56.0 ± 5.52 for DS16. MSNN showed a slight but significant difference between QH34 (13.7 ± 0.7) and DS16 (12.0 ± 0.4) (Figure 1B). The correlation analysis revealed strong positive relationships among traits across both years (Figure 1C). Notable correlations included the following: a strong correlation between PN2023 and SN2023 (r = 0.89), a positive correlation between MSNN2024 and PN2023 (r = 0.63), a high correlation between SN2024 and PN2024 (r = 0.70), and a strong relationship between MSNN2024 and SN2024 (r = 0.70).

Frequency distribution analysis revealed distinct patterns in trait distribution between genotypes DS16 and QH34 across both years (Figure 2). For PN in 2023 (Harbin), the distribution showed a relatively normal pattern with the highest frequency occurring between 40 and 55 pods, encompassing both DS16 and QH34 populations. In 2024 (Sanya), the PN distribution shifted leftward, showing lower values overall, with peak frequencies between 20 and 35 pods, demonstrating the year-over-year reduction in pod number. SN distributions in 2023 displayed a broader range, with frequencies peaking between 80 and 120 seeds, showing a slight positive skew. The 2024 data revealed a compressed distribution with the highest frequencies occurring between 40 and 60 seeds, reflecting the overall reduction in seed production. Both genotypes showed overlapping distributions, though QH34 consistently maintained higher ranges. MSNN showed distinctive distribution patterns. In 2023, the distribution was relatively symmetric, centered between 16 and 22 seeds per node, with the highest frequency around 20 MSNN. The 2024 distribution shifted notably lower, with peak frequencies between 7 and 11 MSNN, demonstrating a more condensed range. This shift aligns with the overall reduction in reproductive traits observed in 2024. The frequency distributions support the statistical findings of decreased trait values in 2024 compared to 2023, while also revealing the nature of trait variation within and between genotypes. The distributions generally showed more positive skewness in 2024, particularly for PN and SN, which aligns with the higher skewness values reported in the statistical analysis (skewness = 1.96 for PN and 2.21 for SN in 2024).

The phenotypic analysis of PN, SN, and MSNN revealed notable variations across 2023 and 2024 (Table 1). In 2023, PN ranged from 22.8 to 104.0 with a mean of 52.37 ± 13.24, while SN exhibited broader variation ranging from 35.6 to 239.4 with a mean of 111.43 ± 34.22. The MSNN varied from 8.0 to 28.8 with an average of 18.57 ± 3.61. In 2024, all traits showed generally lower values compared to the previous year. PN ranged from 9.8 to 82.0 with a mean of 27.25 ± 9.89, representing a substantial decrease from 2023. Similarly, SN showed reduced values ranging from 22.0 to 192.0 with a mean of 57.16 ± 21.35. MSNN also decreased, ranging from 5.2 to 15.8 with an average of 8.98 ± 1.94. The coefficient of variation (CV) remained relatively stable across both years, ranging from 0.22 to 0.37, suggesting consistent relative variability despite the differences in absolute values. Notably, the 2024 data exhibited higher skewness and kurtosis values, particularly for PN (skewness = 1.96, kurtosis = 6.96) and SN (skewness = 2.21, kurtosis = 8.85), indicating a more asymmetric distribution compared to 2023. Variance analysis revealed that seed number consistently showed the highest variation among all traits (1171.23 in 2023 and 455.82 in 2024), while MSNN displayed the lowest variance (13.07 in 2023 and 3.77 in 2024). This pattern suggests that SN is the most variable trait among those measured, while MSNN remains relatively stable.

### 2.2. Construction of the High-Density Genetic Map

Based on the reference genome, a total of 6717 polymorphic markers were located on 20 chromosomes. After filtering out the tags with MLOD values less than 3 with the SLAF tags, a total of 6297 markers were mapped and located as markers. The mapping rate was 93.75%. Each chromosome is a linkage group. The linear arrangement of markers within the linkage group was obtained via HighMap software analysis, and the genetic distance between adjacent markers was estimated. Finally, a high-density genetic map with a total map distance of 2945.26 cM was obtained. First, to test the quality of the soybean high-density genetic map, the basic information of the number of markers, total map distance, average map distance, maximum gap, and gap < 5 cM ratio of each linkage group was statistically analyzed (Table 2). The soybean high-density genetic map contains 6297 SLAF tags, with a total map distance of 2945.26 cM and an average map distance of 0.47 cM. The number of tags on each chromosome ranges from 131 to 625, the map distance ranges from 122.39 to 192.55 cM, the average map distance ranges from 0.24 to 1.37 cM, the ratio of spacing < 5 cM ranges from 97.73% to 100%, and the maximum spacing ranges from 4 to 14.34 cM.

### 2.3. QTL Mapping and Genetic Architecture

QTL mapping identified multiple significant loci controlling these yield-related traits (Figure 3, and Table 3). In Harbin, six QTLs were detected across chromosomes 6, 14, 18, 19, and 20. The most significant QTL was identified for MSNN on chromosome 19 (LOD = 22.59), explaining 24.57% of the phenotypic variation with a positive additive effect of 2.22. Two significant QTLs for SN were detected on chromosomes 14 and 18, collectively explaining 7.35% of the phenotypic variation. A single QTL for PN was identified on chromosome 20, explaining 5.85% of the phenotypic variation. The Sanya analysis revealed four major QTLs, with chromosome 19 emerging as a crucial genomic region. A highly significant MSNN QTL on chromosome 19 (LOD = 37.92) explained 43.59% of the phenotypic variation. This same chromosomal region harbored significant QTLs for both PN (LOD = 18.16, PVE = 23.02%) and SN (LOD = 15.24, PVE = 19.59%), suggesting potential pleiotropy or tight linkage among yield-related traits. An additional MSNN QTL was identified on chromosome 17 (LOD = 3.89), explaining 3.46% of the phenotypic variation. Physical lengths of QTL intervals varied considerably, ranging from 64.58 kb to 5.54 Mb, indicating different levels of mapping resolution. Notably, the major QTL cluster on chromosome 19 was consistently detected across both years, particularly for MSNN, suggesting stable genetic control of these yield-related traits despite environmental variation between years. Additive effect analysis revealed that QH34 alleles generally contributed positively to trait values at major QTLs, particularly those on chromosome 19, consistent with the superior performance of QH34 in phenotypic evaluations. This comprehensive genetic architecture analysis provides valuable insights into the molecular basis of yield-related trait variation in this population.

### 2.4. Gene GO Enrichment Analysis

Through GO enrichment analysis, we found that most of the genes within chr.19 QTL cluster intervals were involved in cellular processes and metabolic processes (Figure 4). Most of the genes were involved in intracellular metabolic processes, bioregulation, and reaction to stimuli, and were mostly involved in binding and catalytic activities in terms of molecular functions. The QTL in chr.19 contained only 17 annotated genes, most of which were involved in growth and cellular processes. and intracellular metabolic processes. GO enrichment and KEGG pathway analyses highlighted key biological processes, cellular components, molecular functions, and pathways associated with differentially expressed genes (DEGs). In the GO enrichment analysis, the figure emphasizes the significance of RNA processing pathways, particularly “mRNA splicing, via spliceosome”, which shows a high -log10 *p*-value (>2), along with other related processes such as “deadenylation-dependent decapping” and “termination of RNA polymerase II transcription”. The enrichment of splicing-related complexes such as U1, U2AF, U4, and U5 snRNPs, as well as the SMN–Sm protein complex, further supports the importance of RNA splicing in the study. Molecular functions such as RNA-binding and nucleosidase activity are also highlighted, reflecting their roles in RNA metabolism. The KEGG pathway analysis reinforces these findings, with the spliceosome pathway showing the highest enrichment score (~8, *p*-value < 0.1). Additionally, pathways related to aminoacyl-tRNA biosynthesis, RNA degradation, and mRNA surveillance were significantly enriched, further underscoring the centrality of RNA-related processes. Metabolic pathways, including starch and sucrose metabolism (enrichment score ~8), O-glycan biosynthesis, nicotinate and nicotinamide metabolism, galactose metabolism, and pyrimidine metabolism, were also prominently featured, suggesting extensive metabolic reprogramming. Signaling pathways such as MAPK signaling and plant hormone signal transduction, though modestly enriched, were statistically significant, indicating their potential roles in cellular responses. 

### 2.5. Candidate Genes Mining

Seventeen candidate genes of interest were pinpointed within the chr.19 QTL region, each with distinct functional roles (Appendix A). Glyma.19G191600 encodes a Serine/threonine-protein kinase CTR1, which is involved in ethylene signaling and stress responses, potentially regulating plant growth and development under stress conditions. Glyma.19G193100 encodes Serine/threonine-protein kinase KIPK1, associated with abiotic stress signaling pathways. Glyma.19G194800 encodes Cell division protein ftsz homolog 2–1, playing a role in cell division and chloroplast organization. Glyma.19G195300 encodes Kinesin-like protein KIN-5C, involved in intracellular transport and cell division, suggesting a role in cellular dynamics. Glyma.19G196000 encodes Probable UDP-N-acetylglucosamine--peptide N-acetylglucosaminyltransferase SPINDLY, which regulates growth through protein glycosylation. Glyma.19G196300 encodes an mRNA-decapping enzyme-like protein, involved in mRNA turnover and post-transcriptional regulation. Glyma.19G199100 encodes a protein involved in damaged DNA binding and DNA-directed DNA polymerases, likely playing a role in DNA repair and replication. Glyma.19G199400 encodes a BTB/POZ domain-containing protein, potentially involved in protein–protein interactions and transcriptional regulation. Glyma.19G199900 encodes an Aluminium-activated malate transporter family protein, which may contribute to aluminum tolerance and organic acid transport. Glyma.19G200300 encodes a Dof-type zinc finger DNA-binding family protein, a transcription factor that may regulate gene expression in response to environmental or developmental cues. Glyma.19G200400 encodes a Tetratricopeptide repeat (TPR)-like superfamily protein, likely involved in protein-protein interactions and cellular signaling. Glyma.19G200800 encodes nuclear factor Y, subunit A10, a component of a transcription factor complex that regulates gene expression. Glyma.19G200900 encodes a Glutaredoxin family protein, involved in redox regulation and stress responses. Glyma.19G201100 encodes Ubiquitin-specific protease 8, which regulates protein stability and function through deubiquitination. Glyma.19G201200 encodes Dihydroneopterin aldolase, involved in the biosynthesis of tetrahydrobiopterin, a cofactor for enzymatic reactions. Glyma.19G201300 has no annotation provided, requiring further analysis to determine its function. Finally, Glyma.19G201400 encodes Calmodulin-domain protein kinase CDPK isoform 2, involved in calcium signaling and stress responses. These genes collectively highlight diverse roles in stress responses, growth regulation, DNA repair, metabolism, and gene expression, providing a strong foundation for further functional studies.

### 2.6. Differential Gene Expression Patterns Between QH34 and DS16 Lines

Figure 5 shows the gene expression patterns, which reveal distinct developmental responses in QH34 and DS16. Analysis of gene expression patterns across developmental stages (V1-V3, R1-R2) revealed distinct transcriptional responses between QH34 and DS16 genotypes (Figure 5a,b). We observed several notable expression patterns among the seventeen examined genes from chromosome 19. In QH34, *Glyma.19G201300* and *Glyma.19G201400* exhibited dramatic upregulation during the reproductive stage R1, with expression levels increasing approximately 500-fold and 750-fold, respectively, compared to vegetative stages. This sharp increase suggests these genes play crucial roles in the transition to reproductive development. Notably, *Glyma.19G194800* showed similar temporal dynamics, with peak expression (34.44-fold) coinciding with the R1 stage (Figure 5b). DS16 displayed markedly different expression patterns (Figure 5a). The most striking observation was the substantial upregulation of *Glyma.19G199400* during the R2 stage (41.16-fold increase), while this gene maintained relatively low expression levels in QH34 across all stages (Figure 5d). Similarly, *Glyma.19G201300* reached its expression peak (50.75-fold) during R1 in DS16, showing a distinct temporal pattern compared to QH34. Several genes demonstrated stage-specific expression patterns common to both genotypes. For instance, *Glyma.19G200900* showed elevated expression during early vegetative stages (V1-V2) in both genotypes, with higher levels in DS16 (4.68-fold) compared to QH34 (2.70-fold). Conversely, Glyma.19G200400 maintained relatively stable expression across all developmental stages in both genotypes, suggesting a housekeeping role. Comparative analysis between genotypes revealed that DS16 generally exhibited higher expression amplitudes for most genes during reproductive stages (Figure 5c). This was particularly evident for *Glyma.19G193100*, which showed a 16.85-fold increase during R2 in DS16 compared to only 0.75-fold in QH34. These genotype-specific differences suggest distinct regulatory mechanisms governing developmental transitions. A subset of genes (*Glyma.19G200800, Glyma.19G201100*, and *Glyma.19G196000*) showed moderate expression levels throughout development in both genotypes, with slight increases during reproductive stages. This pattern indicates their potential involvement in both vegetative and reproductive development, albeit with less stage-specific regulation. Interestingly, we observed coordinated expression patterns among several gene clusters. For example, *Glyma.19G199100, Glyma.19G200300*, and *Glyma.19G201200* showed synchronized expression peaks during V3-R1 stages in both genotypes, suggesting possible co-regulation or involvement in related developmental processes. These findings highlight the complex transcriptional networks governing developmental transitions in both genotypes and reveal potential key regulators of reproductive development. The distinct expression patterns between QH34 and DS16 provide valuable insights into genotype-specific developmental regulation and may explain observed phenotypic differences between these lines.

## 3. Discussion

Complex quantitative traits in soybeans are generally interrelated [45], and influenced by genetic and environmental variables [18]. The efficacy and precision of QTL mapping for quantitative traits may be considerably influenced by its application in a particular context. This study performed QTL mapping of three yield traits, including PN, MSNN, and SN, in two locations in China to alleviate environmental influences. Furthermore, varying QTL mapping results are often observed across different approaches because of the distinct assumptions and models employed, each containing unique advantages and disadvantages. This research utilized QTL mapping by SLAF-seq to construct a detailed genetic framework. QTL mapping is an effective method for examining quantitative traits in plants [46,47]. The quality of genetic maps is essential for the precision of QTL mapping. Studies indicate that augmenting the number of genetic markers can improve the resolution of genetic maps [48,49]. Nonetheless, soybeans demonstrate elevated levels of linkage disequilibrium (LD) relative to other plants [50]. In our study, we utilized SLAF-seq for extensive marker identification and genotyping to construct a high-density genomic map. SLAF-seq has numerous benefits, such as efficient marker development, cost efficiency, and capacity for managing extensive populations. The soybean high-density genetic map contains 6297 SLAF tags, with a total map distance of 2945.26 cM and an average map distance of 0.47 cM. The number of tags on each chromosome ranges from 131 to 625, the map distance ranges from 122.39 to 192.55 cM, the average map distance ranges from 0.24 to 1.37 cM, the ratio of spacing <5 cM ranges from 97.73% to 100%, and the maximum spacing ranges from 4 to 14.34 cM. This soybean genetic map exhibited significantly enhanced marker density and diminished inter-marker distances compared to earlier maps. Consequently, SLAF fragment length selection is not plagued by random interruptions and thus achieves better repeatability over RAD-seq and GBS. Moreover, large amounts of sequence information can be generated and whole-genome density distributions can be handled using SLAF-seq [51] to generate markers with higher density, better consistency, greater effectiveness, and at lower cost than using traditional methods.

The phenotypic analysis demonstrated significant genotypic variation between DS16 and GH34, with GH34 consistently showing superior performance across traits. In Harbin, GH34 exhibited substantially higher values for all traits (PN: 77.0 vs. 45.2; SN: 156.0 vs. 96.4; MSNN: 22.5 vs. 16.8), indicating its genetic potential for yield improvement. The pronounced environmental variation, particularly the reduced trait values in Sanya, highlights the strong environmental influence on these yield components. This environmental sensitivity is evidenced by the substantial decreases in trait means from Harbin to Sanya. Similar environmental effects on soybean yield components have been reported by Zhang et al. [19], who found significant year-to-year variations in pod and seed numbers across different genetic backgrounds. The most significant finding of our study is the identification of a major QTL cluster on chromosome 19 that consistently influenced all three traits across both environments. This genomic region demonstrated remarkable stability and effect size, explaining up to 43.59% of MSNN variation in Sanya. The co-localization of QTLs for PN (PVE = 23.02%) and SN (PVE = 19.59%) in this same region suggests possible pleiotropy or tight linkage. This finding aligns with previous studies by Li et al. [21], who identified similar QTL clusters affecting multiple yield components in soybeans. The physical length of QTL intervals varied considerably, with the chromosome 19 cluster showing particularly high mapping resolution, comparable to recent high-density mapping studies by Wang et al. [24].

Plant pod and seed production involves complex genetic regulation across multiple biological processes [52]. The final numbers of pods and seeds depend on several key factors: the number of reproductive organs (both male and female), the viability of gametophytes, and how the zygote develops after fertilization [53]. In this research, the genes found within the major QTL regions on chromosome 19 appear to be associated with reproductive development. Similar genes in other crop species are known to regulate various developmental processes, including protein ubiquitination, cell division, growth, and proliferation, petal differentiation, auxin response, gibberellin signaling, and plant self-incompatibility [54,55]. Notably, there was significant overlap in the known genes controlling PN, SN, and MSNN within these QTL regions. However, determining the specific genes responsible for these traits based solely on functional annotations remains challenging. Gene ontology analysis of the chromosome 19 QTL cluster revealed enrichment in cellular processes and metabolic pathways, particularly RNA-processing and -splicing mechanisms. The identification of 17 candidate genes within this region provides specific molecular targets for functional validation. Key candidates include serine/threonine protein kinases and cell division proteins, similar to those identified in yield-related studies by Chen et al. [20]. The differential expression patterns between GH34 and DS16 across developmental stages reveal distinct regulatory mechanisms, with some genes showing dramatic upregulation during reproductive stages. These expression patterns align with findings from Liu et al. [47], who reported stage-specific regulation of yield-related genes in soybeans. The stable QTL cluster on chromosome 19, combined with the identified candidate genes and their expression patterns, provides multiple avenues for yield improvement. The strong environmental effects observed suggest that breeding strategies should focus on developing varieties with both high yield potential and stability across environments, as suggested by recent studies [2,4,30]. The co-localization of QTLs for multiple traits offers opportunities for simultaneous improvement of yield components through marker-assisted breeding. The integration of advanced sequencing technologies, robust statistical methods, and functional validation strategies in this study sets a new benchmark for QTL mapping and candidate gene identification in soybeans. Our findings not only provide novel insights into the genetic architecture of PN, SN, and MSNN but also offer valuable targets for marker-assisted breeding to improve soybean yield and plant architecture.

## 4. Materials and Methods

### 4.1. Soybean Populations Assessments

The soybean RILs were developed from two parents, ‘Qihuang 34′, and ‘Dongsheng 16′. A population of 325 individual F_2:6_ plants was created through single-seed descent. In 2023, the parents and the RIL populations were planted at the Harbin Experimental Farm, and in 2024 planted at the Yazhou District Experimental Farm in Sanya. Each treatment for the two parents and the RILs was replicated three times in a randomized block design, with plots measuring 3.0 m in row length, 35 cm spacing between rows, 10 cm spacing between plants, and three rows per plot. On 30 April each year, seeds were hand sown, placing two seeds in each hole. If both seeds germinated, one plant was removed after it produced its second set of trifoliate leaves. The entire field was irrigated with 120 m^3^ of water before sowing. Following seed emergence, field management practices adhered to local agricultural guidelines.

### 4.2. Phenotypic Statistics and Analysis

Ten mature plants from each plot were selected randomly in the middle of each row to measure MSNN, PN, and SN before the harvest in the field for each replication. The average of the three replications was used for phenotypic data analysis. Descriptive statistics, linear regression analysis, and calculation of the skewness and the kurtosis were performed on phenotypic data from two environments using GraphPad Prism software (GraphPad Software Inc. v.7.0.4, La Jolla, CA, USA), and the corresponding tables or figures were generated.

### 4.3. DNA Extraction, Construction and Genotyping of SLAF Libraries

Fresh leaves from the two parents and all 325 RILs were collected into centrifuge tubes, frozen in liquid nitrogen, ground in a tissue grinder, and then stored at −80 °C. Total genomic DNA was extracted from each leaf sample using a modified CTAB method [56]. The quality and concentration of the extracted DNA were determined by 1% agarose gel electrophoresis and a spectrophotometer (UV-Vis Spectrophotometer Q5000/Waltham, MA, USA). Individual SLAF libraries of 325 RILs and their parents were constructed and sequenced. A combination of Rsa I and Hae III restriction endonucleases was selected to digest genomic DNA. The SLAF markers of 325 RILs were grouped and genotyped using the method of Sun et al. [41]. The high-density molecular tags developed for the RIL population were used to construct a genetic map using HighMap software (http://highmap.biomarker.com.cn (accessed on 2 March 2025)) [57].

### 4.4. SLAF-Sequence and Genotype Parental Lines and RILs

The sequencing depth of Dongsheng 16 was 13.17××, the sequencing depth of Qihuang 34 was 12.77×, the average sequencing depth of parents was 12.97×, and the average sequencing depth of offspring was 10.05×. A total of 627,099 SLAF tags were obtained. Among them, there were 98,571 polymorphic SLAF tags, and the polymorphism ratio reached 15.72%. The SLAF tags were located on the reference genome using BWA software (https://nchc.dl.sourceforge.net/project/bio-bwa/bwa-0.7.15.tar.bz2 (accessed on 2 March 2025)) [58,59], and the chromosome distribution map of SLAF tags (Figure 6a) and polymorphic SLAF tags (Figure 1B) were drawn according to the distribution of SLAF tags on chromosomes.

### 4.5. Polymorphic SLAF Tag Encoding and Screening of Mapping SLAF Tags

To facilitate subsequent genetic analysis, all polymorphic SLAF tags need to be genotyped. The genotype encoding rule is the common two-allele encoding rule in genetics. This study is a RIL segregation population, so aa x bb type polymorphic tags were selected, and the paternal genotype was aa, the maternal genotype was bb, and the offspring genotype was aa (paternal type), bb (maternal type), ab (heterozygous type), and -(missing type). Based on the results of parental genotype detection, sites with missing parental information were filtered out. According to the genotype encoding rules in the above table, the 98,571 polymorphic SLAF tags obtained in this study were typed, and 35,174 tags were successfully encoded. To ensure the quality of the genetic map, the polymorphic SLAF tags were filtered to obtain 6717 SLAF tags that can be used for mapping. The distribution of the chromosome is shown in Appendix A.

### 4.6. QTL Mapping Analysis of RIL Population

QTL IciMapping V4.2 software was used for QTL detection based on the phenotypes in the two environments. The complete interval mapping method (ICIM-ADD) was used, the scanning step was set to 1 cM, the stepwise regression marker entry probability was 0.001, and LOD = 3.0 was used as the threshold to detect the existence of a putative QTL. In addition, the ICIM-EPI method was used to detect epistatic interactions between markers, with a threshold of 5.0.

### 4.7. Candidate Gene Prediction

Gene ontology (GO) term enrichment analysis was conducted for genes located within the identified QTL clusters. This step aids in selecting candidate genes by examining sequence variations between the parental lines. GO annotation data were retrieved from the soybean database SoyBase (https://www.soybase.org/ (accessed on 28 December 2024)), and the WEGO2.0 online tool (http://wego.genomics.org.cn/ (accessed on 28 December 2024)) was utilized to generate visualizations of gene enrichment across cellular components, molecular functions, and biological processes. Subsequently, genes exhibiting high expression in soybeans were further screened using expression data obtained from SoyBase. Functional annotation of the selected genes was performed using the Phytozome database (https://phytozome-next.jgi.doe.gov/ (accessed on 28 December 2024)), and candidate genes were identified based on literature reviews and gene function analyses. Tissue-specific expression patterns of the candidate genes were visualized using a heat map generated by the online tool (https://www.omicshare.com/tools/Home/Soft/heatmap/ (accessed on 28 December 2024)) [60], while gene structures were mapped using the Gene Structure Display Server (http://gsds.gao-lab.org/index.php/ (accessed on 28 December 2024)). Additionally, resequencing data of the parental lines were analyzed to compare variations in candidate genes. The whole genomes of the parental lines were sequenced using the Illumina HiSeq X Ten platform, achieving an average sequencing depth of 8× [61]. High-quality sequencing data were evaluated to predict structural variations in the candidate genes.

### 4.8. Quantitative Real-Time PCR (qRT-PCR) Analysis

Quantitative real-time PCR (qRT-PCR) was employed to analyze gene expression levels. Root and stem tissues from the two parental lines, DS16 and QX34, were stored at –80 °C. Total RNA was extracted from these tissues using the Plant Total RNA Purification Kit (Promega (Beijing) Biotech Co., Ltd., Beijing, China). Following extraction, 1 µg of RNA was treated to remove genomic DNA and then reverse transcribed into cDNA using the TransScript One-Step gDNA Removal and cDNA Synthesis SuperMix kit (Novoprotein). The qRT-PCR was performed on the CFX96 Real-Time System (Bio-Rad, Hercules, CA, USA) with the following cycling conditions: initial denaturation at 95 °C for 1 minute, followed by 40 cycles of denaturation at 95 °C for 10 s, annealing at 55.0–60.0 °C (gene-specific) for 10 s, and extension at 72 °C for 30 s. The soybean ACT3 gene served as the reference control. Expression levels were quantified using the comparative cycle threshold method (2^−△△ct^), and each experiment was replicated three times. Primer sequences for the target genes are provided in Appendix A.

## 5. Conclusions

This study provides a comprehensive analysis of the genetic and molecular mechanisms underlying yield-related traits in soybeans, focusing on PN, SN, and MSNN. Phenotypic analysis revealed significant variations between genotypes DS16 and QH34, with QH34 consistently exhibiting higher trait values, though environmental effects were evident across growing seasons. A high-density genetic map enabled the identification of stable QTLs, particularly on chromosome 19, which explained up to 43.59% of the phenotypic variation for MSNN and harbored key candidate genes involved in stress responses, growth regulation, and metabolism. Gene expression analysis highlighted genotype-specific transcriptional responses, with QH34 showing dramatic upregulation of reproductive-stage genes, suggesting distinct regulatory mechanisms.

## Figures and Tables

**Figure 1 ijms-26-02300-f001:**
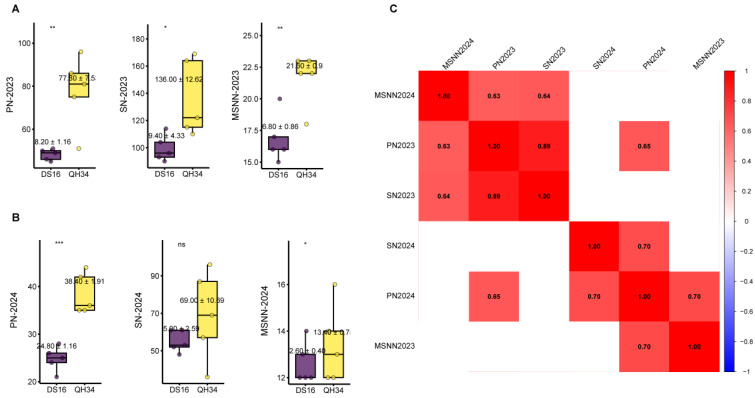
Phenotypic variation and correlation analysis of yield-related traits in soybean genotypes DS16 and QH34 across two growing seasons. (**A**) Comparison of pod number (PN), seed number (SN), and main stem node number (MSNN) between DS16 and QH34 genotypes in 2023. Bar plots show mean values ± standard error (*n* = 5). (**B**) Comparison of PN, SN, and MSNN between DS16 and QH34 genotypes in 2024. Boxplots show mean values ± standard error (*n* = 5). (**C**) Correlation matrix showing Pearson’s correlation coefficients (r) among yield-related traits across both growing seasons. Color intensity indicates correlation strength, with red representing positive correlations. Significant differences between genotypes: * *p* < 0.05, ** *p* < 0.01, *** *p* < 0.001; ns: not significant.

**Figure 2 ijms-26-02300-f002:**
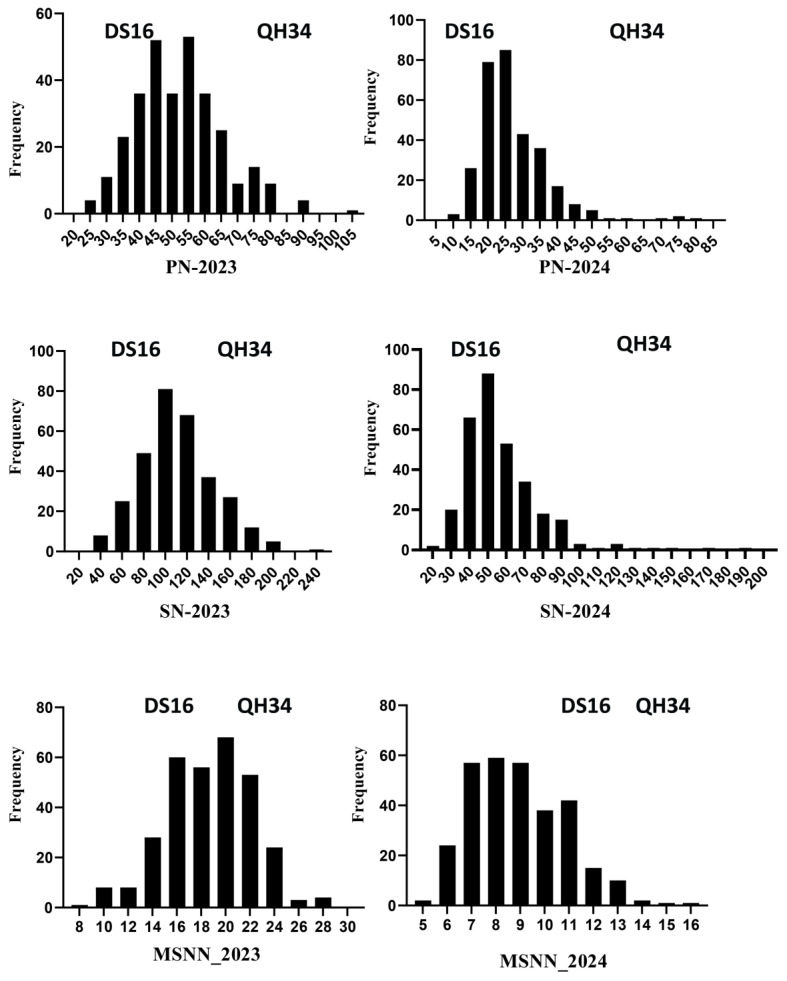
Frequency distribution of PN, SN, and MSNN for DS16 and QH34 genotypes in Harbin and Sanya.

**Figure 3 ijms-26-02300-f003:**
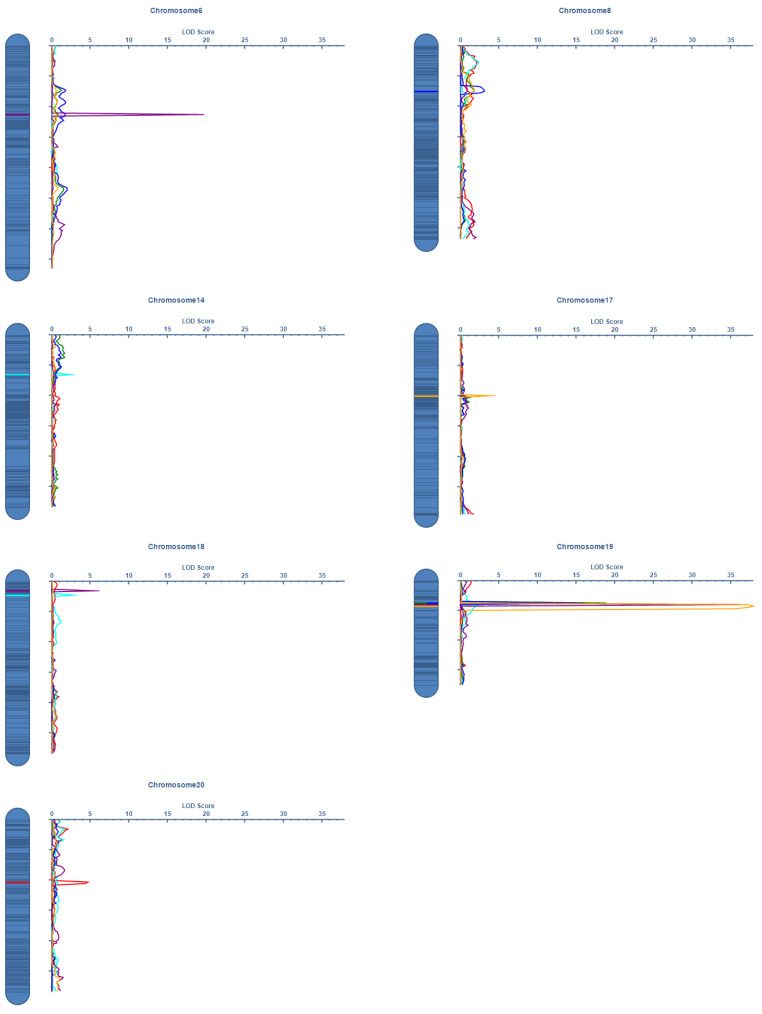
The QTLs of PN, SN, and MSNN traits identified in the RIL-F_2:6_ population using a high-density genetic map. The x-axis indicates genetic position (cM); the y-axis indicates LOD score. Different colored lines represent different traits and locations.

**Figure 4 ijms-26-02300-f004:**
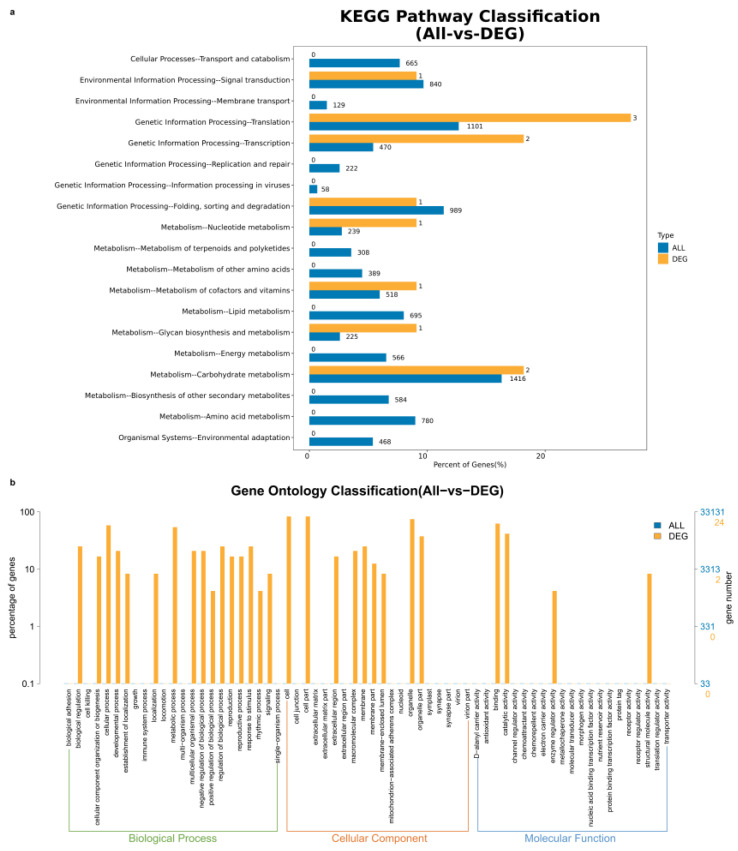
Enrichment analysis of candidate genes. (**a**) KEGG pathway analyses, (**b**) GO enrichment analysis.

**Figure 5 ijms-26-02300-f005:**
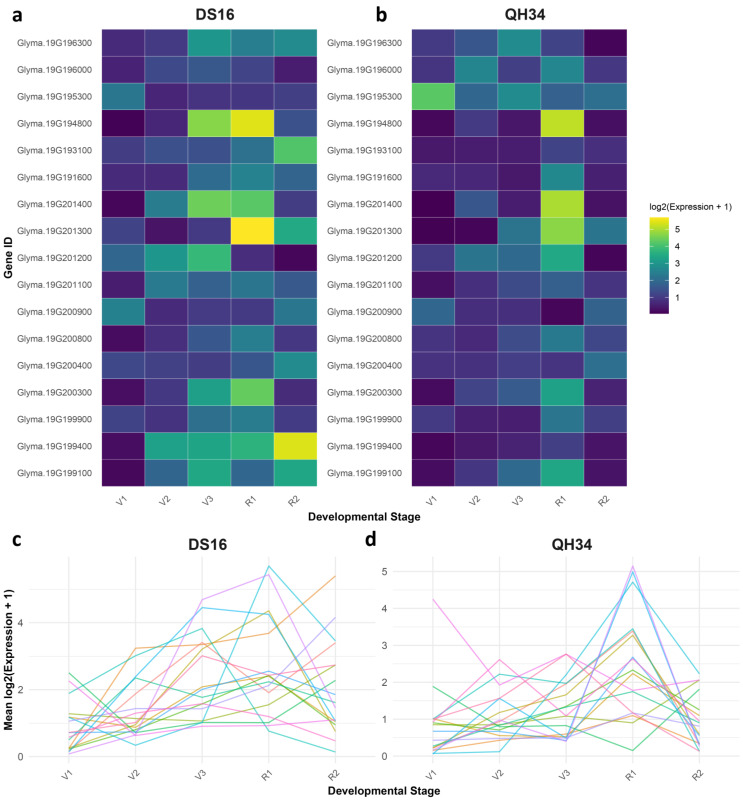
Expression profiles of 17 selected genes in the soybean parental lines DS16 and QH34 across various developmental stages. (**a**) and (**b**) display heatmaps representing the expression levels in DS16 and QH34, respectively, with rows corresponding to individual genes and columns indicating developmental stages. (**c**,**d**) show line graphs summarizing the mean expression levels of these genes across developmental stages for DS16 and QH34.

**Figure 6 ijms-26-02300-f006:**
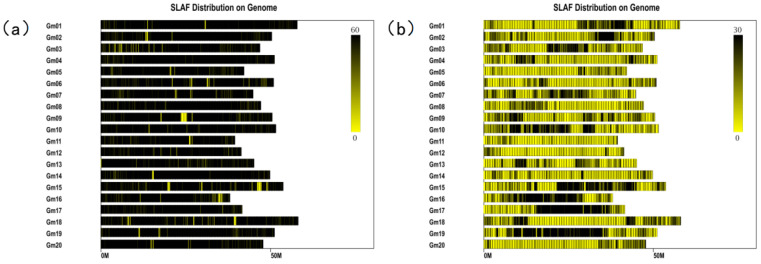
SLAF tag and polymorphism distribution of SLAF tag on chromosomes. (**a**) shows the distribution of SLAF markers, and (**b**) shows the polymorphic distribution of SLAF markers (in M units). Note: The abscissa is the length of a chromosome, and each yellow stripe represents a chromosome. The genome is divided into 1M units, and different colors indicate the number of SLAF. The darker the color, the more SLAF. The darker areas in the figure are the concentrated distribution of SLAF markers. Note: The horizontal axis is the length of the chromosome, and each yellow stripe represents a chromosome.

**Table 1 ijms-26-02300-t001:** Statistical analysis of phenotypic traits for PN, SN, and MSNN across two environments.

Environment	Traits	Min	Max	Mean	SD	Variance	Skewness	Kurtosis	CV
Harbin	PN	22.8	104	52.37	13.24	175.39	0.55	0.45	0.25
	SN	35.6	239.4	111.43	34.22	1171.23	0.4	0.19	0.31
	MSNN	8	28.8	18.57	3.61	13.07	−0.2031	−0.0508	0.23
Sanya	PN	9.8	82	27.25	9.89	97.76	1.96	6.96	0.36
	SN	22	192	57.16	21.35	455.82	2.21	8.85	0.37
	MSNN	5.2	15.8	8.98	1.94	3.77	0.524	−0.1229	0.22

**Table 2 ijms-26-02300-t002:** Basic information statistics of the high-density genetic map.

Linkage Group	Marker (No)	Map Distance (cM)	Average Map Distance (cM)	Gaps < 5 cM (%)	Max Gap (cM)
Gm01	348	145.08	0.42	100.00	4.54
Gm02	372	156.70	0.42	99.46	5.56
Gm03	365	167.92	0.46	99.18	7.46
Gm04	309	140.07	0.45	99.03	7.28
Gm05	183	160.41	0.88	98.35	5.38
Gm06	211	145.65	0.69	99.05	8.32
Gm07	371	161.22	0.44	99.46	9.49
Gm08	255	125.88	0.50	100.00	4.00
Gm09	357	133.83	0.38	99.72	5.50
Gm10	455	123.76	0.27	99.56	7.94
Gm11	131	122.39	0.94	96.92	10.95
Gm12	133	128.10	0.97	97.73	14.34
Gm13	243	125.73	0.52	99.17	10.69
Gm14	224	148.46	0.67	99.55	11.70
Gm15	625	167.77	0.27	99.36	7.90
Gm16	390	192.55	0.49	99.23	7.58
Gm17	345	148.67	0.43	98.84	6.76
Gm18	283	142.60	0.51	100.00	4.43
Gm19	484	152.16	0.32	99.79	6.31
Gm20	213	156.31	0.74	99.06	5.82
Total	6297	2945.26	0.47	99.17	14.34

**Table 3 ijms-26-02300-t003:** QTL mapping identified multiple significant loci in soybeans.

Year	Traits	Chr.	Position	Left Marker	Right Marker	LOD	PVE(%)	Add	Physical Length
Harbin	PN	20	41	Marker2448322	Marker2501150	4.6804	5.8529	−3.3846	83,365 bp
	SN	14	26	Marker5556237	Marker5717653	2.5198	3.4514	−6.1588	36.05 kbp
	SN	18	9	Marker4230716	Marker4407522	2.8469	3.8975	−6.5107	381 kbp
	MSNN	6	45	Marker777950	Marker766908	7.8568	7.7379	−1.2729	5.54 Mbp
	MSNN	18	6	Marker4165022	Marker4228378	5.493	3.804	−0.7547	64.58 kbp
	MSNN	19	16	Marker1939109	Marker1978745	22.5907	24.5668	2.2155	101.90 kbp
Sanya	PN	19	15	Marker1990755	Marker1770324	18.1568	23.0188	4.6921	274,223 bp
	SN	19	15	Marker1990755	Marker1770324	15.2414	19.5888	9.4462	274,223 bp
	MSNN	17	40	Marker5907974	Marker5935243	3.8931	3.4569	−0.3523	2.84 Mbp
	MSNN	19	17	Marker1978745	Marker1711920	37.9218	43.588	1.2453	1.12 Mbp

## Data Availability

Data is contained within the article and Appendix A.

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
