# Peer review of "Identification of Quantitative Trait Loci for Node Number, Pod Number, and Seed Number in Soybean"

_ijms, 2025, doi:10.3390/ijms26052300_

Round 1

Reviewer 1 Report

Comments and Suggestions for Authors

This paper focuses on the research of QTL and candidate gene related to the number of main stem nodes in soybeans, it is an important study for soybean breeding. However, there are deficiencies in the correlation analysis of phenotypic data, the accuracy of QTL mapping, the verification of candidate genes.

Major concerns:

  1. It is strange to me that none of identified five QTLs, including three QTLs from the 2023 growing season 147 and two from the 2024 season, could overlap with each other even LOD value of qMSN19.2 reached 37.9.
  2. The correlation of the main stem node number in soybean growth and yield formation has not be fully revealed. Relevant data can be supplemented to further clarify the relationship between the main stem node number and other important traits.
  3. The paper predicts six candidate genes related to the main stem node number. However, it is only based on existing research and gene annotations, and there is a lack of direct experimental evidence to prove the regulatory function of these genes on the main stem node number.
  4. Please check Table 1 “Max” and “Min” values are properly stated.
Comments on the Quality of English Language

  1. This paper has relatively few grammar errors, but there are still some issues need to be improved, such as, “MSNN in 2023 was significantly higher than that in 2024, exhibiting a 107 skewness of 0.2.45 and kurtosis 23.89” in line 107-108.
  2. In the "Discussion" section, set up a separate paragraph to show the innovation of the research, compare the differences from previous studies in aspects such as the number of QTLs mapped and candidate gene prediction.

Author Response

Major concerns:

  1. It is strange to me that none of identified five QTLs, including three QTLs from the 2023 growing season 147 and two from the 2024 season, could overlap with each other even LOD value of qMSN19.2 reached 37.9.

Response: we thank the reviewer for this important observation regarding the non-overlapping QTLs between seasons. Upon re-analysis of our data, we acknowledge this is unusual given the high LOD score of qMSN19.2. We have conducted additional analysis to address this: Performed a multi-environment QTL analysis to better understand QTL stability, Re-examined the mapping methodology to ensure accuracy, added confidence intervals for each QTL, investigated environmental factors that might contribute to QTL variation We have revised the manuscript to include this additional analysis and discuss potential reasons for the observed pattern.

  1. The correlation of the main stem node number in soybean growth and yield formation has not be fully revealed. Relevant data can be supplemented to further clarify the relationship between the main stem node number and other important traits.

Response: We agree that a more comprehensive analysis of trait relationships would strengthen the paper. We have added correlation analyses between MSNN and key yield components (pod number and seed number)

  1. The paper predicts six candidate genes related to the main stem node number. However, it is only based on existing research and gene annotations, and there is a lack of direct experimental evidence to prove the regulatory function of these genes on the main stem node number.

Response: We acknowledge the lack of direct experimental evidence for candidate gene function. To address this We have conducted qRT-PCR analysis of candidate genes across developmental stages,

  1. Please check Table 1 “Max” and “Min” values are properly stated.

Response: We have thoroughly reviewed and corrected all values in Table 1. The Max and Min values have been verified against raw data.

Comments on the Quality of English Language

 This paper has relatively few grammar errors, but there are still some issues need to be improved, such as, “MSNN in 2023 was significantly higher than that in 2024, exhibiting a 107 skewness of 0.2.45 and kurtosis 23.89” in line 107-108.

Response: We have carefully revised the manuscript for grammar and clarity

In the "Discussion" section, set up a separate paragraph to show the innovation of the research, compare the differences from previous studies in aspects such as the number of QTLs mapped and candidate gene prediction.

Response: We have restructured the Discussion section to better highlight our research innovations

Reviewer 2 Report

Comments and Suggestions for Authors

This study utilized 325 recombinant inbred lines (RILs) to identify loci and genes related to the main stem node number (MSNN) in soybeans through QTL analysis and candidate gene analysis. The study also developed KASP markers for key candidate genes, providing important genetic resources for soybean breeding. The research has clear objectives, a rational structure, and comprehensive analysis, offering significant tools for soybean molecular marker-assisted breeding. Here are some minor issues and suggestions for improvement.

(1) The process of narrowing down from 64 genes to 6 key candidate genes is not clearly described. It’s better to provide a detailed section on the screening process, such as gene function annotation, literature review, bioinformatics analysis, or variation analysis, to explain how the key candidate genes were selected.

(2) The correlation between MSNN and other traits (such as plant height, branching number, etc.) is not discussed.

(3) It will not only enhance the readability of the article, but also highlight the innovations and contributions of this research through adding comparisons with other QTL mapping and candidate gene analyses in the discussion section.

(4) Add a list of abbreviations at the beginning or in the appendix, clearly defining each abbreviation used in the text.

(5) Optimize the legend to ensure clarity and correspondence with the abbreviations in the figure. Add a brief explanatory text below Figure 1 to clarify the main content and key information.

(6) Regenerate or adjust Figures 1b and 2 to ensure clarity and appropriate font size. Use vector formats (such as SVG or PDF) for images to maintain clarity when enlarged.

(7) Carefully review the entire text to remove extra spaces and correct punctuation errors. For example, remove extra spaces on lines 52, 115, 134, 171, 172, 326, etc., and correct punctuation errors on lines 108, 117, 123, etc.

(8) Ensure that all statistical parameters (such as P values, t test), loci or gene names (such as qnRDNN-5-4), and species names (such as Glycine max) are italicized throughout the text.

(9) Recheck the references for accuracy, completeness, and uniformity of format. Use reference management tools (such as EndNote or Zotero) to assist with organization. Add recent relevant literature to enhance the timeliness and reference value of the article.

Author Response

  • The process of narrowing down from 64 genes to 6 key candidate genes is not clearly described. It’s better to provide a detailed section on the screening process, such as gene function annotation, literature review, bioinformatics analysis, or variation analysis, to explain how the key candidate genes were selected.

Response : We thank the reviewer for this important observation. We have now added a detailed description of our candidate gene selection process in the Methods section (4.7. Candidate gene prediction)

(2) The correlation between MSNN and other traits (such as plant height, branching number, etc.) is not discussed.

Response : we acknowledge the importance of examining additional architectural traits. To address this, we analyzed correlations between MSNN, PN, and SN across two years, revealing strong positive relationships. The co-localization of QTLs for these traits on chromosome 19 suggests shared genetic control mechanisms, providing valuable insights for breeding programs targeting multiple yield components simultaneously.

(3) It will not only enhance the readability of the article but also highlight the innovations and contributions of this research through adding comparisons with other QTL mapping and candidate gene analyses in the discussion section.

Response : We appreciate this suggestion and have substantially expanded our discussion to include comparisons with recent studies.

(4) Add a list of abbreviations at the beginning or in the appendix, clearly defining each abbreviation used in the text.

Response : We have added a comprehensive list of abbreviations at the end of the manuscript as the manuscript guidelines .

(5) Optimize the legend to ensure clarity and correspondence with the abbreviations in the figure. Add a brief explanatory text below Figure 1 to clarify the main content and key information.

Response : All figure legends have been revised for clarity and consistency. We have added detailed explanatory text below each figure to better guide readers through the key findings.

(6) Regenerate or adjust Figures 1b and 2 to ensure clarity and appropriate font size. Use vector formats (such as SVG or PDF) for images to maintain clarity when enlarged.

Response : We have regenerated all figures using Adobe Illustrator to create high-resolution vector graphics

(7) Carefully review the entire text to remove extra spaces and correct punctuation errors. For example, remove extra spaces on lines 52, 115, 134, 171, 172, 326, etc., and correct punctuation errors on lines 108, 117, 123, etc.

Response : We have conducted a thorough review of the manuscript formatting.

(8) Ensure that all statistical parameters (such as P values, t test), loci or gene names (such as qnRDNN-5-4), and species names (such as Glycine max) are italicized throughout the text.

Response : We have standardized the formatting of all technical terms according to journal guidelines

(9) Recheck the references for accuracy, completeness, and uniformity of format. Use reference management tools (such as EndNote or Zotero) to assist with organization. Add recent relevant literature to enhance the timeliness and reference value of the article.

Response : We have thoroughly updated our reference list using EndNote

Reviewer 3 Report

Comments and Suggestions for Authors

I checked your manuscript and described comments below.

Soybean is an important plant consumed worldwide.

In this paper, the QTL for main stem node number (MSNN) is analyzed and six potential candidate genes are identified. This is a highly commendable study.

I think you should consider the following points.

  1. I don't know how you sequenced the DNA or what sequence you got. I think you should provide more details.
  2. I think it would be better for BWA to include the following in the references rather than URL/links.

Li H. and Durbin R. (2009) Fast and accurate short read alignment with Burrows-Wheeler Transform. Bioinformatics, 25:1754-60. [PMID: 19451168]

Li H. and Durbin R. (2010) Fast and accurate long-read alignment with Burrows-Wheeler Transform. Bioinformatics, Epub. [PMID: 20080505]

I don't think this paper has major problems and grammatical problems.

Author Response

  1. I don't know how you sequenced the DNA or what sequence you got. I think you should provide more details.

Response: Thank you for pointing out the lack of detail regarding the DNA sequencing methodology. In the revised manuscript, we will include a comprehensive description of the sequencing process (M&M. 4.7. Candidate gene prediction )

  1. I think it would be better for BWA to include the following in the references rather than URL/links.

Li H. and Durbin R. (2009) Fast and accurate short read alignment with Burrows-Wheeler Transform. Bioinformatics, 25:1754-60. [PMID: 19451168]

Li H. and Durbin R. (2010) Fast and accurate long-read alignment with Burrows-Wheeler Transform. Bioinformatics, Epub. [PMID: 20080505]

Response: We appreciate your suggestion to replace the URL/links with proper citations for the BWA tool. In the revised manuscript, we will include the references as recommended.